# Metabolic Pathways Affected in Patients Undergoing Hemodialysis and Their Relationship with Inflammation

**DOI:** 10.3390/ijms25179364

**Published:** 2024-08-29

**Authors:** María Peris-Fernández, Marta Isabel Roca-Marugán, Julià L. Amengual, Ángel Balaguer-Timor, Iris Viejo-Boyano, Amparo Soldevila-Orient, Ramon Devesa-Such, Pilar Sánchez-Pérez, Julio Hernández-Jaras

**Affiliations:** 1Health Research Institute Hospital La Fe, 46026 Valencia, Spain; maria_peris@iislafe.es (M.P.-F.); marta_roca@iislafe.es (M.I.R.-M.); 2University and Polytechnic La Fe Hospital, 46026 Valencia, Spain; ivb_1993@hotmail.com (I.V.-B.); amsolde@gmail.com (A.S.-O.); devesasuch@hotmail.com (R.D.-S.); sanchez_pil@gva.es (P.S.-P.); 3Big Data AI and Biostatistics Platform, Health Research Institute Hospital La Fe, 46026 Valencia, Spain; julian_amengual@iislafe.es (J.L.A.); angel_balaguer@iislafe.es (Á.B.-T.)

**Keywords:** uremic toxins and inflammation, metabolic pathways affected in patients undergoing renal replacement therapies

## Abstract

Worldwide, 3.9 million individuals rely on kidney replacement therapy. They experience heightened susceptibility to cardiovascular diseases and mortality, alongside an increased risk of infections and malignancies, with inflammation being key to explaining this intensified risk. This study utilized semi-targeted metabolomics to explore novel metabolic pathways related to inflammation in this population. We collected pre- and post-session blood samples of patients who had already undergone one year of chronic hemodialysis and used liquid chromatography and high-resolution mass spectrometry to perform a metabolomic analysis. Afterwards, we employed both univariate (Mann–Whitney test) and multivariate (logistic regression with LASSO regularization) to identify metabolites associated with inflammation. In the univariate analysis, indole-3-acetaldehyde, 2-ketobutyric acid, and urocanic acid showed statistically significant decreases in median concentrations in the presence of inflammation. In the multivariate analysis, metabolites positively associated with inflammation included allantoin, taurodeoxycholic acid, norepinephrine, pyroglutamic acid, and L-hydroorotic acid. Conversely, metabolites showing negative associations with inflammation included benzoic acid, indole-3-acetaldehyde, methionine, citrulline, alphaketoglutarate, n-acetyl-ornithine, and 3-4-dihydroxibenzeneacetic acid. Non-inflamed patients exhibit preserved autophagy and reduced mitochondrial dysfunction. Understanding inflammation in this group hinges on the metabolism of arginine and the urea cycle. Additionally, the microbiota, particularly uricase-producing bacteria and those metabolizing tryptophan, play critical roles.

## 1. Introduction

Chronic kidney disease (CKD) is an escalating global health threat with an estimated 850 million individuals living with the disease in 2017 and it being the third fastest-growing cause of death [1]. Of those individuals, approximately 3.9 million globally rely on kidney replacement therapy (KRT) indefinitely [1], predominantly through hemodialysis. They face an elevated risk of cardiovascular illness and death and are more prone to infections and malignancies [2]. For instance, in the age group of 40–44 years, there is a notable gap in life expectancy between men undergoing dialysis (10.9 years) and men in the general population (36.5 years), exceeding 25 years (over 30 years for women) [1]. Cardiovascular disease (CVD) impacts over two-thirds of individuals undergoing hemodialysis (HD), representing a significant source of morbidity and contributing to nearly half of all mortality cases [3].

Inflammation serves as a crucial link between chronic kidney disease and cardiovascular issues and is related to the prognosis of this group of patients. For example, inflammation directly results in hemodynamic strain, elevating ventricular wall pressure and initiating myocardial stun, and prolonged exposure ultimately culminates in irreversible myocardial injury [4]. Furthermore, uremic inflammation is mechanistically linked to biological aging through processes like telomere shortening, oxidative stress, impaired nutrient sensing, and systematic inflammation, which collectively contribute to cellular stress and vascular damage [5,6].

Inflammation in end-stage renal disease (ESRD) patients undergoing KRT is triggered by a combination of intrinsic and extrinsic factors. The dialysis process itself, including exposure to dialysis membranes and the presence of central venous catheters or peritoneal dialysis (PD) catheters, introduces foreign elements that provoke an immune response. Additionally, fluid overload resulting from sodium and water retention creates a pro-inflammatory state, further aggravating cardiovascular strain in both HD and PD patients [7]. This is compounded by internal factors, such as dysbiosis, an imbalance in gut microbiota, which leads to the accumulation of uremic toxins. These toxins, along with oxidative stress driven by mitochondrial dysfunction and cellular senescence, perpetuate a cycle of inflammation that is difficult to interrupt [8].

Despites advances in understanding the role of medium-sized-molecule toxins in this inflammatory cascade, much remains to be uncovered about the complex metabolic pathways involved. The interplay between these toxins and other molecular players in CKD and cardiovascular disease is an area that should be explored [9].

To do this, metabolomics emerges as a promising avenue for unraveling novel pathways within the broader field of “omics” sciences. This discipline employs diverse analytical techniques to identify and quantify small molecules (metabolites) in biological samples, offering insights to the individual phenotype as it accounts for both the genome and the extrinsic factors, such as lifestyle, medications, and underlying health conditions. This makes it a particularly suitable tool to assess uremic toxins in the body.

Untargeted metabolomics, which focuses on the comprehensive detection and relative quantification of metabolites, but can make the exact identification of metabolites difficult, has been utilized to investigate the changes in the metabolic profile in hemodialysis patients before and after sessions and comparing different hemodialysis techniques [10,11,12]. Meanwhile, targeted metabolomics is used to perform a precise measurement of predefined, much smaller groups of metabolites [13].

A review of recent literature using the terms “metabolomics” AND “hemodialysis” AND “inflammation” revealed 18 studies on this topic. We excluded those that did not use metabolomics or focused on non-dialysis patients, leaving nine key studies. Notable findings include the following: Holle et al. (2022) on gut microbiota and inflammation [14]; Kim et al. (2021) comparing dialyzer types [15]; a 2022 study linking endocabinoids and inflammation [13]; Tsalik et al. on metabolite markers in critically ill patients [16]; TMAO’s association with inflammation in peritoneal dialysis and hemodialysis [17,18]; AKI studies identifying inflammatory proteins and metabolites [19,20]; and glycomics research on PD complications [21]. Our work is unique in using semi-targeted metabolomics and LASSO to create a predictive model for distinguishing inflamed from non-inflamed patients.

We use an in-house polar compound library to facilitate the identification of measured metabolites within a unified workflow, which serves as a bridge between untargeted and targeted procedures. This allows us to broaden the search scope for potential metabolites related to inflammation while accurately identifying them

In this way, we aim to characterize the metabolic profile within this cohort, with a particular emphasis on inflammation-associated metabolites as potential therapeutic targets.

## 2. Results

The study group, with an average age of 68.97 years (SD ± 14.18), comprised 37% females and showed an average height of 165.23 cm (SD ± 8.87) and a mean wight of 75.03 kg (SD ± 16.73). Approximately 54% of the patients were smokers, and an equivalent percentage had type 2 diabetes. Hypertension was prevalent in 93% of the cohort, while dyslipidemia was present in 79%.

The summary of the demographic and clinical characteristics of the sample (*n* = 43) can be found in Table 1.

A semi-targeted metabolic strategy was utilized, employing liquid chromatography and high-resolution mass spectrometry alongside an in-house library of polar compounds. The identified metabolites and their mass-to-charge ratios are available in the Appendix A. Both univariate (Mann–Whitney test) and multivariate (logistic regression with LASSO (Least Absolute Shrinkage and Selection Operator) regularization) analyses were conducted to identify metabolomic variables linked to inflammation.

The analyzed patients had already undergone one year of hemodialysis, and we obtained samples before and after a session to try to identify which metabolites change with the procedure.

A numerical summary of the pre-session samples is provided the Table 2, including the respective medians and interquartile ranges of the variables with significant changes between the two groups, and Figure 1 visually represents this information in a volcano plot.

In the pre-session samples, the median concentrations of indole-3-acetaldehyde (*p* = 0.0023), 2-ketobutyric acid (*p* = 0.0196), and urocanic acid (*p* = 0.0323) significantly decreased in the presence of inflammation, as indicated by a statistically significant Mann–Whitney test. The magnitude of change was greater in carnitine, which saw a decrease in the median concentration by a factor of 1.39 in inflamed patients, and lowest in trehalose, with a decrease by only a 0.75-factor.

Table 3 reports the standardized coefficients (SCs) and odds ratios (ORs) associated with each metabolite, and Figure 2 visually represents the ORs obtained in the logistic-LASSO model.

Regarding this multivariate analysis, metabolites showing positive associations with inflammation (inflammation = yes) include allantoin (SC = 1.17, OR 3.22), taurodeoxycholic acid (SC = 0.43, OR = 1.54), norepinephrine (SC = 0.4, OR 1.49), pyroglutamic acid (SC = 0.38, OR = 1.46), and L-hydroorotic acid (SC = 0.29, OR = 1.34), whereas benzoic acid (SC = −0.97, OR 0.38), indole-3-acetaldehyde (SC = −0.74, OR 0.48), methionine (SC = −0.71, OR 0.49), citrulline (SC = −0.60, OR 0.55), alpha-ketoglutarate (SC = −0.55, OR = 0.55), n-acetyl-l-ornithine (SC = −0.53, OR 0.59), and 3-4-dihydroxybenzeneacetic acid (SC = −0.40, OR 0.67) show negative associations. Additionally, n-acetylcarnitine (SC = −0.17, OR 0.85), 3-indolepropionic acid (SC = −0.17, OR 0.85), pantothenate (SC = −0.16, OR 0.85), and cholic acid (SC = −0.15, OR 0.86) exhibit minor negative influences. Odds ratios close to 1 indicate variables with a lesser impact on the model.

A numerical summary of the results obtained in the post-session samples is provided the Table 4, including the respective medians and interquartile ranges of the variables showing significant changes between the two groups, and Figure 3 visually represents this information in a volcano plot.

Regarding this, univariate analysis revealed significant variables that also exhibited a substantial change in the magnitude of their medians: m-coumaric acid showed higher median levels in patients with inflammation compared to those without, while n-acetyllysine and aminoadipic acid had lower median levels in patients with inflammation.

However, when we performed the multivariate analysis in the post-session samples it failed to satisfactorily discriminate between patients with inflammation and those without inflammation, possibly due to the removal of a large portion of metabolites during the hemodialysis technique.

## 3. Discussion

Our objective was to characterize the metabolic profile related to inflammation, defined by PCR > 2 ng/mL, in dialysis-dependent patients one year before and after undergoing a hemodialysis session.

In the pre-session group, we identified indole-3-acetaldehyde as a significant variable associated with a lack of inflammation in both univariate and multivariate analyses.

This compound, a product of tryptophan metabolism by gut microbiota has been shown to modulate immune responses, including in inflammatory bowel diseases, metabolic syndrome, and cardiovascular health [22]. Interestingly, while indole-3-acetaldehyde is linked to pro-inflammatory effects in other contexts, our findings suggest a protective role in hemodialysis patients, aligning with its broader influence on immune regulation [23,24].

In the univariate analysis, the variables identified as associated with a lack of inflammation were 2-ketobutyric acid and urocanic acid.

2-Ketobutyric acid, involved in amino acid and propionate metabolism, increases NAD+ levels and autophagy, promoting longevity and cellular health [25,26]. Urocanic acid, derived from histidine, functions as an immunomodulator, particularly in skin UV protection an immune response regulation. Although its role in hemodialysis patients is not well documented, our study suggests that it may contribute to reduced inflammation [25,27,28].

In contrast, the multivariate analysis identified allantoin, taurodeoxycholic acid, norepinephrine, and L-hydroorotic acid as markers of an inflammatory profile.

Allantoin, a marker of oxidative stress, has been linked to increased mortality in chronic kidney disease, consistent with our findings of its association with inflammation [29,30].

Taurodeoxycholic acid, a bile acid, may reflect a physiological response to inflammation particularly in the context of cardiovascular dysfunction. The exogenous administration of taurine-conjugated bile acids has been linked to increased vasodilation by enhancing nitric oxide (NO) production from the endothelium and thus offers promising strategies for treating endothelial dysfunction in cardiovascular disease. In a study published in 2020, it was found that in individuals with type 2 diabetes mellitus (T2DM), there is a significant increase in the levels of this molecule compared to age- and gender-matched individuals without T2DM. Consequently, it is possible that the endogenous production of this acid could be related to the physiological response to inflammation and could function as a biomarker in these types of patients, although its exogenous administration has beneficial properties [31,32].

Norepinephrine, a recognized uremic toxin, has been implicated in chronic inflammation and may exacerbate the inflammatory state in hemodialysis patients. It also serves as a primary transmitter in the majority of postganglionic sympathetic nerve fibers. The overactivity of the sympathetic nervous system contributes to the development and the extension of several inflammatory conditions [33,34].

Pyroglutamic acid, linked to glutathione depletion and oxidative stress, is elevated in conditions like sepsis, supporting its role as a marker of inflammation. Additionally, a study published in 2024, which explored the causal relationship between gut microbial taxa, human blood metabolites, and serum inflammatory markers, revealed that pyroglutamic acid contributes to elevated IL-6 and SAA1 levels [35]. Considering the impaired renal excretion and chronic inflammation common in dialysis patients, it is logical for them to exhibit elevated levels of pyroglutamic acid [36,37,38].

L-hydroorotic acid, involved in pyrimidine synthesis, may be upregulated due to the overactivity of the CAD complex, contributing to the inflammatory response [39,40,41]. Moreover, inhibiting this complex proved to be beneficial in inflammatory situations, such as COVID infection [42] or certain types of cancer [43].

Furthermore, the analysis highlighted benzoic acid, methionine, citrulline, alphaketoglutarate, and N-acetyl-L-ornithine as potential markers for identifying non-inflamed patients.

Benzoic acid (BA) is naturally present, both in its free form and as esters in many plant and animal species. Appreciable amounts have been found in most berries. BA ingestion has been found to increase nutrient digestibility and to decrease ammonia nitrogen levels in the gut. Furthermore, it enhances mucosal integrity and promotes intercellular junction formation, in addition to influencing immunological responses and the microbiota [44]. Hippuric acid is the excreted form of benzoic acid. Benzoic acid undergoes metabolism through butyrate-CoA ligase, producing an intermediate compound called benzoyl-CoA, which is subsequently metabolized by glycine N-acyltransferase to form hippuric acid. The protection that appears to be provided by the presence of benzoic acid could also be related to the decrease in activity of this pathway, which would result in lower levels of hippuric acid. Hippuric acid in advanced CKD has been associated with atherogenesis through the disruption of the endothelial function and renal fibrosis [45,46,47].

Methionine, an indispensable amino acid containing sulfur, plays a crucial role in synthesizing key molecules, including antioxidants like gluthatione, along non-essential amino acids, such as carnitine and cysteine, as well as methyl donors, like S-adenosyl methionine and vasodilatory intermediates. Eggs, sesame seeds, fish, meat, and cereal grains are among the primary sources of methionine [48]. Methionine supplementation can modulate inflammation and oxidative stress reducing the expression of inflammatory and fibrogenic genes while enhancing endogenous antioxidant capacity and protecting against oxidative damage [49,50,51].

Citrulline is an important metabolite in arginine metabolism. Asymmetric dimethylarginine (ADMA) is a metabolite derived from arginine and can inhibit nitric oxide synthase (NOS), thereby limiting the production of nitric oxide and potentially leading to vascular complications associated with end-stage renal disease. ADMA is primarily metabolized by dimethylarginine dimethylaminohydrolases (DDAH-1 and DDAH-2) and converted into dimethylamine and citrulline. Patients with kidney dysfunction can have lower concentrations of DDAH-1, which is particularly abundant in the kidney, therefore impeding the further metabolization of citrulline in the arginine–NO pathway. Elevated levels of ADMA are associated with cardiovascular disease, hypertension, and immune dysfunction [52]. Therefore, we could argue that patients with increased citrulline have better preserved metabolism of arginine and nitric oxide, and thus, the presence of this metabolite is related to the absence of inflammation. Other studies focusing on patients performing hemodialysis have also found higher citrulline levels [13]. This metabolite has been shown to be significantly reduced after hemodialysis session in adults with ERSD, which may signal impaired NO synthesis post-procedure [53]. That is possibly the reason why the levels of this metabolite were not significantly associated with the absence of inflammation in the post-hemodialysis examination, as its levels have been reduced to nearly zero. Additionally, citrulline levels can be increased with the exogenous administration of L-carnitine in patients undergoing chronic hemodialysis [54].

Alpha-ketoglutarate (AKG) is an important intermediate of various metabolic pathways, including the tricarboxylic acid (TCA) cycle, anabolic and catabolic reactions of amino acids, and collagen biosynthesis [55]. Recently there have been several reported health-promoting effects of AKG, including anti-aging, extension of a healthy lifespan, anti-osteoporotic, anti-inflammatory, and anticarcinogenic effects, as well as protection against pressure overload cardiomyopathy. These beneficial effects are not only metabolic but also mediated through epigenetic mechanisms [56]. Specifically, it has been recognized as an antioxidant agent, enhancing the activity of other antioxidant enzymes, such as SOD, CAT, and gluthation peroxidase. Other antioxidant mechanisms of AKG include the decomposition of hydrogen peroxide and the promotion of mitophagy to clear damaged mitochondria [55]. However, it is important to note that recent research has shown that AKG can induce ROS through autophagy induction [57]. AKG is widely considered safe and bioavailable, making the oral or intravenous supplementation of AKG an effective therapeutic approach [55]. Additionally, SGLT inhibitors have been shown to increase renal levels of AKG, which may be another factor to explain their cardiovascular benefits [58].

In summary, it makes sense that AKG is associated with reduced inflammation in hemodialysis patients, and furthermore, it suggests a potential therapeutic target in these individuals.

It is remarkable that although alpha-ketoglutarate can be formed from several amino acids one of the pathways starts with histidine. Histidine is an essential amino acid that is obtained from meat and other high-protein products. Histidine is oxidized and deaminated to urocanic acid by the enzyme histidase. Urocanic acid is subsequently hydrolyzed to form N-formiminoglutamate (FIGu). This compound binds to tetrahydrofolate, releasing glutamate. Finally, glutamate is degraded to alpha-ketoglutarate [59].

Both urocanic acid and alpha-ketoglutarate help us to distinguish the group of non-inflamed patients, suggesting that the activation of this metabolic pathway may confer some protection to patients on hemodialysis. At this stage of kidney disease, the protein needs of patients increase to 1.3 g/kg/day, so changes in the diet could help to explain these observations. In Figure 4, we observe the relationship of urocanic acid, alphaketoglutarate, and 2-oxybutyrate with the Krebs cycle, the increase in autophagy, and mitochondrial biogenesis and their connection to the reduction of the inflammatory environment.

Within the ornithine cycle, ornithine serves as the direct substrate, and n-acetylornithine is a N-acetylation derivative, which can also serve as a precursor of ornithine [25]. The ornithine cycle is a pivotal amino acid metabolic pathway linked to various diseases, such as infection [60], cancer [61], and hypertension [62]. A study published in 2021 found that n-acetyl-ornithine was positively correlated with multiple cytokines and coagulation parameters in COVID-19 patients, especially those with severe cases [63]. Currently, there are no cases in the literature that associate this metabolite with protective characteristics against inflammation. However, other studies suggest that oxidative stress upregulates the transcription of enzymes, such as ornithine decarboxylase, which produces an increase in polyamines, such as spermine and spermidine [64]. Diabetic rats have lower levels of spermine and spermidine, which cause heart damage and inflammation, leading to fibrosis. Spermidine treatment showed promising results in reversing this harmful process [65]. Another study found that ornithine, through its conversion to putrescine via Arg 1 and ornithine decarboxylase, plays a critical role in enhancing continual efferocytosis by macrophages. This is shown graphically in Figure 5. This metabolic pathway supports the optimal clearance of apoptotic cells and promotes the resolution of tissue injury [66]. Therefore, the fact that N-acetylornithine levels are associated with the absence of inflammation could indicate a more conserved polyamine metabolic pathway and, consequently, a better response to inflammation.

In a 2022 study, 3-4-dihydroxybenzeneacetic acid, also known as 3,4-dihydroxyphenylacetic acid or DOPAC/DHPAA, resulted in the partial reversal of pro-inflammatory microRNA responses in human intestinal epithelial cells exposed in cranberry extract [67]. Conversely, another study published in 2020 highlights the anti-inflammatory potential of epicatechin and dihydroxybenzoic acid from cocoa in alleviating diabetic kidney inflammation through targeted cellular pathways, contrasting the lack of similar effects observed with 3,4-dihydroxyphenylacetic acid [68].

Finally, DOPAC/DHPAA is a metabolite of dopamine, and its reduction indicates dopamine dysfunction in Parkinson’s disease. In an article published in 2023, it was found that empagliflozin treatment restored these metabolite levels, which suggests the potential stabilization of dopaminergic functions [69].

In the post-session analysis, m-coumaric acid was elevated in inflamed patients, while n-acetyllysine and aminoadipic acid were lower. M-coumaric acid’s pro-oxidative properties align with inflammation [70,71], whereas n-acetyllysine’s role in genome maintenance and repair might suggest a protective effect [72,73]. Aminoadipic acid, although linked to metabolic dysfunction [74], did not show significant differentiation in the multivariate analysis, highlighting the challenges of analyzing metabolite concentrations post-dialysis.

In comparison with our previous study analyzing inflammation-associated metabolites in stage 5 non-dialysis-dependent CKD patients, we continue to observe the significance of mitochondrial energy metabolism. However, in this case, patients with preserved metabolic pathways that promote autophagy and mitochondrial biogenesis appear to be more protected against inflammation. Additionally, the metabolism of arginine, the urea cycle, and the microbiota, which interact with several metabolic pathways, seem to play an important role [75].

This study has several limitations. Firstly, our data only indicate trends or changes in response intensities, rather than precise concentrations. Secondly, we opted to use an in-house library of polar compounds to streamline results and interpretation, which may have led to the omission of some metabolites. Lastly, due to time and budget constraints, we were limited in our sample size, restricting the scope for multiple comparisons and subgroup analysis, particularly regarding the cause of kidney disease.

Due to these limitations, future research should expand these findings to validate identified biomarkers and explore their clinical utility in larger patient cohorts, potentially paving the way for personalized therapeutic interventions aimed at mitigating inflammation in hemodialysis patients.

## 4. Materials and Methods

### 4.1. Experimental Design and Cohort

Our objective was to investigate the influence of various metabolites on inflammation in patients who had undergone one year of chronic hemodialysis. We conducted an observational prospective, single-center study, where we gathered the patient’s clinical and analytical data and information about the hemodialysis sessions. The sample size consisted of 43 patients, with 23 patients in the “inflammation no” group and 20 patients in the “inflammation yes” group.

Initially, we conducted a descriptive analysis of the study population to characterize its demographics. Subsequently, we explored the metabolomic profiles of the patients before and after the hemodialysis session through the use of liquid chromatography and high-resolution mass spectrometry.

The inclusion criteria comprised an age over 18 years and the patient having undergone hemodialysis sessions for more than a one year at the Hospital La Fe or at a peripheral dialysis center affiliated with it. The exclusion criteria included suboptimal initiation due to clinical instability (defined as acute decompensation requiring hospitalization, requiring temporal CVC insertion, or an unexpected and unwanted change in techniques).

We collected data about age, gender, height, weight, BMI, cardiovascular risk factors, urea, uric acid, corrected calcium for serum albumin, phosphate, sodium, potassium, chloride, CRP, hemoglobin, leucocytes, and platelets. Regarding the hemodialysis session, we collected data on the type of technique employed, the access used, the residual diuresis at the start of the procedure, the average interdialytic weight gain, and the Kt/V to ensure efficacy.

### 4.2. Metabolomics Analysis

The processing of the samples was similar to that conducted in the article published by our group this year [75].

The serum centrifuged from the samples was collected and stored at −80 °C, and subsequently, a procedure for protein precipitation and the extraction of polar compounds was performed by adding 180 µL of cold methanol to 20 µL of the serum. After a double centrifugation at 13,000× *g* (10 min, 4 °C), 20 µL of the supernatant was transferred to a 96-well plate for liquid chromatography coupled to mass spectrometry analysis. Additionally, 70 µL of water and 10 µL of the standard mix solution (reserpine, leucine, enkephalin, phenylanalanine-d5, 20 µM) were added to each sample. Quality control samples were prepared by mixing 10 µL from each serum sample. Blank samples were prepared by replacing serum with ultrapure water to identify potential artifacts. Finally, plasma samples, QCs, and blanks were injected into the chromographic system. To minimize intrabatch variability, there was a random injection order and an analysis of quality samples every six plasma samples, with a blank analysis conducted at the end of the sequence.

Ultra-performance liquid chromatography–high-resolution mass spectrometry (UPLC-HRMS) analysis was conducted using and Orbitrap QExactive spectrometer (Thermofisher, Waltham, MA, USA) couple with a ultra-performance liquid chromatography (UPLC) system. Chromatographic separation utilized an Xbridge BEH Amide column (150 × 2 mm, 2.5 µL particle size; Waters, Milford, MA, USA) with a runtime of 25 min, an injection volume of 5 µL, and a column temperature set a 25 °C. The autosampler was maintained at 4 °C, and a flow rate of 105 µL/min was achieved using water and 10 mM ammonium acetate as Mobile Phase A and acetonitrile (ACN) as Mobile Phase B. The gradient protocol included various proportions of Mobile Phase B, starting at 90% and gradually decreasing to 0% over the course of 25 min.

Electrospray ionization was utilized in both positive and negative modes (ESI+/−) for full mass acquisition, with a resolving power of 140,000. Two events were employed: one covering mass ranges from 70 to 700 Da and another covering mass ranges from 700 to 1700 Da. Data were acquired in centroid mode.

Data preprocessing involved converting raw data to mzXML format using a mass converter, followed by processing in EI-MAVEN software (V0.11.0) to generate a peak table containing *m/z* values, retention times, and intensities of polar compounds. Peak areas were extracted and annotated using an in-house polar compound library. Data from positive and negative modes were combined for statistical analysis.

Before conducting statistical analysis, data quality was assessed by evaluating internal standard stability and coefficients of variation (CVs) of quality control samples. Molecular features with a CV > 30% were excluded from the data matrix, and a normalization method (LOESS) was applied to mitigate intrabatch variability resulting from technical differences. The filtered peak table was then used for subsequent statistical analysis.

### 4.3. Statistical Analysis

Patients with CRP values ≤ 2 mg/L were categorized as non-inflamed, while those with CRP > 2 mg/L were considered inflamed. We determined this value cut point based on the identification of CRP levels between 1 and 3 mg/L as the grey zone of inflammation, where cardiovascular risk could possibly increase [76]. All patients underwent a system-based medical and basic physical examination before the sample examination to ensure that the influence of clearly infectious events was minimal.

Initially, we conducted a univariate analysis to explore which metabolomic variables correlated with the presence of inflammation. This involved identifying metabolites significantly upregulated with a fold-change of 1.5 in the inflammatory group compared to the non-inflammatory group using a non-parametric Mann–Whitney test (log2(foldchange) = log2(MInflammation = 1/Minflammation = 0).

For the multivariate analysis, we employed a logistic regression model, suited for binary response variables, applying LASSO regularization. LASSO (Least Absolute Shrinkage and Selection Operator) is a regularization technique aiding in variable selection and regularization by shrinking some regression coefficients towards zero, thus facilitating a more interpretable model. This algorithm optimized the coefficients of logistic regression for each variable (standardized) to enhance performance, specifically reducing classification errors. This was achieved by progressively diminishing coefficients using a multiplier parameter in the algorithm’s cost function until they reached zero, effectively eliminating the corresponding variable. The extent of coefficient minimization and elimination was determined using an additional parameter called lambda. A higher lambda value led to a greater coefficient penalization, reducing their values to zero, and consequently, reducing the number of variables included in the model. The optimal lambda value was determined using a leave-one-out cross-validation. Various models were fitted, with one group of subjects excluded in each iteration to estimate their classification into the “inflamed = no” or “inflamed = yes” group. The error for each case was calculated, and the lambda value minimizing this error was considered the optimal choice. In this model, k was set to 50, implying that one patient was excluded for estimation. In each iteration, there was a focus on minimizing classification errors. Standardized coefficients and odds ratios were computed for each variable. ORs represent the multiplier for the probability of belonging to the inflammation = yes group compared to inflammation no assuming a one standard deviation increase in the variable, as we used standardized coefficients.

The sign of the coefficients indicates how changes in variables affect the probability of inflammation. Positive coefficients imply a higher likelihood of inflammation, while negative coefficients decrease the probability. Variables with larger absolute standardized coefficients hold higher impact in the model, as reflected by their odds ratios (OR = e^coefficient^).

## 5. Conclusions

Our study introduces novel findings through the application of semi-targeted metabolomics and the LASSO statistical method, enabling the creation of a predictive model distinguishing inflamed hemodialysis patients. The metabolism of arginine and therefore the urea cycle are pivotal to the understanding of inflammation in this group of patients.

These results allow us to identify metabolic pathways related to inflammation in hemodialysis patients, thereby identifying potential therapeutic targets and points of optimization in renal replacement therapy.

## Figures and Tables

**Figure 1 ijms-25-09364-f001:**
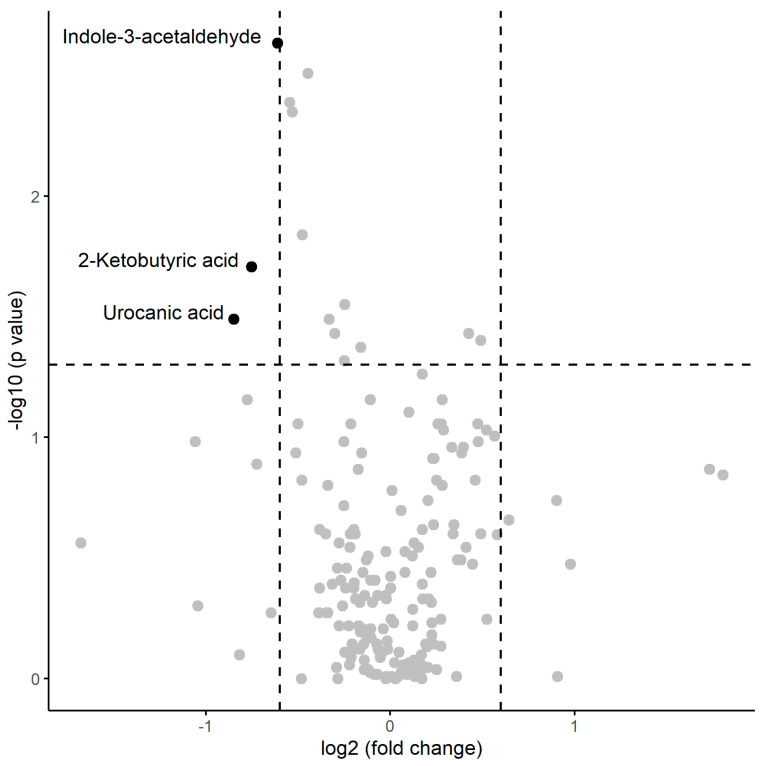
Volcano plot of metabolomic variables. The horizontal dashed line represents the limit of statistical significance (corresponding to a *p*-value = 0.05); values above it denote variables with significantly different medians. The vertical dashed lines represent a log2(fold-change) of −0.6 and +0.6. Values above +0.6 represent variables with a median for inflammation = yes that is 1.5 times in greater magnitude than for inflammation = no. Values below −0.6 represent variables with a median for inflammation = no that is 1.5 times greater in magnitude than for inflammation = yes.

**Figure 2 ijms-25-09364-f002:**
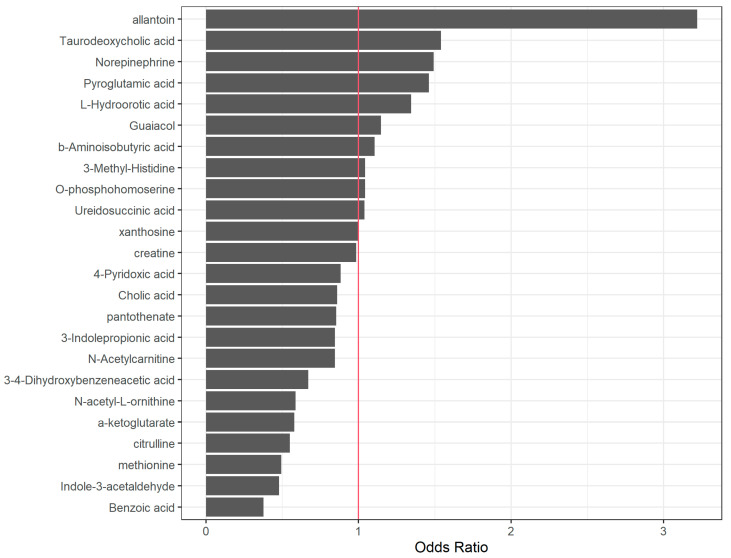
Graphical depiction of the ORs obtained in the Logistic-LASSO model. ORs close to 1 (red vertical line) indicate variables with a lesser impact on the model.

**Figure 3 ijms-25-09364-f003:**
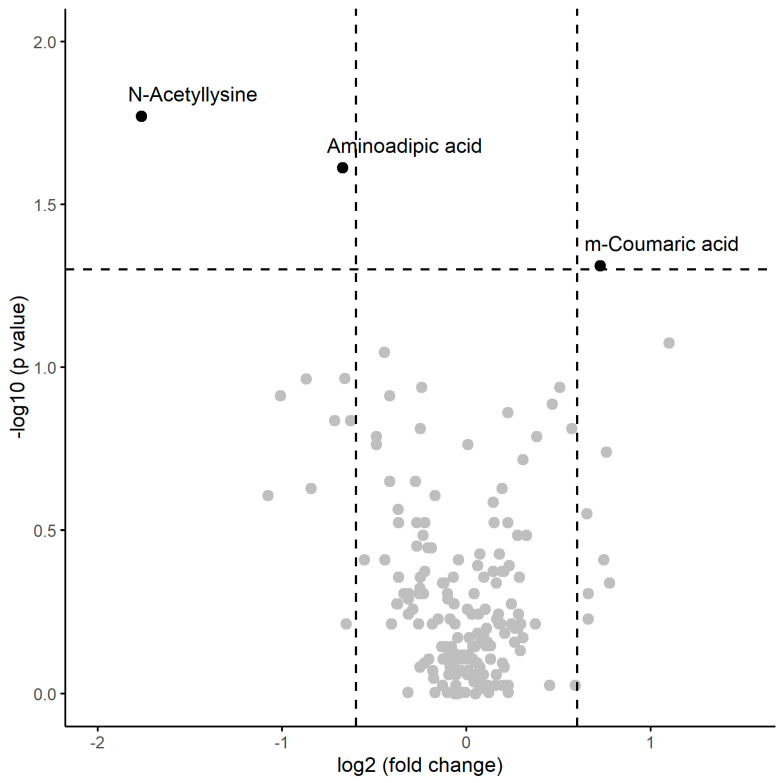
Volcano plot of metabolomic variables. The horizontal dashed line represents the limit of statistical significance (corresponding to *p*-value = 0.05); values above it denote variables with significantly different medians. The vertical dashed lines represent a log2(fold-change) of −0.6 and +0.6. Values above +0.6 represent variables with a median for inflammation = yes that is 1.5 times greater in magnitude than for inflammation = no. Values below −0.6 represent variables with a median for inflammation = no that is 1.5 times greater in magnitude than for inflammation = yes.

**Figure 4 ijms-25-09364-f004:**
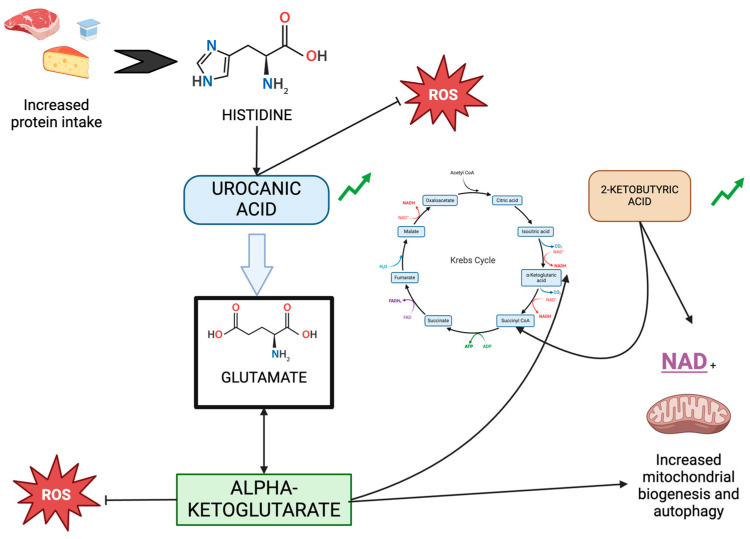
Relationship of urocanic acid, alpha-ketoglutarate, and 2-oxobutyrate with the Krebs cycle, autophagy, and mitochondrial biogenesis.

**Figure 5 ijms-25-09364-f005:**
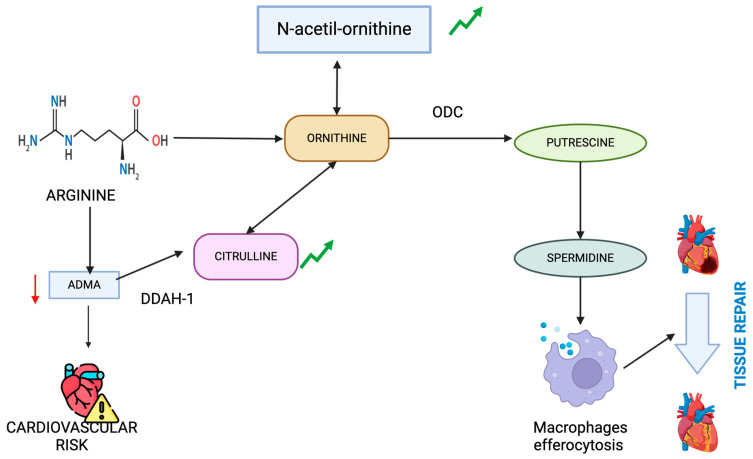
Metabolic pathways involving citrulline and n-acetylornithine, highlighting their roles in arginine metabolism and polyamine synthesis influencing inflammation and tissue health.

**Table 1 ijms-25-09364-t001:** Demographic characteristics of the sample.

Variable	Mean/Frequency
Age (years) (mean ± SD)	68.97 ± 14.18
Female gender, *n* (%)	16 (37%)
Height (cm) (mean ± SD)	165.23 ± 8.87
Weight (kg) (mean ± SD)	75.03 ± 16.73
BMI (mean ± SD)	27.75 ± 4.76
Smoker, *n* (%)	23 (53.5%)
Type 2 diabetes, *n* (%)	21 (48.8%)
Hypertension, *n* (%)	40 (93%)
Dyslipidemia, *n* (%)	34 (79.1%)
Urea (mg/dL) (mean ± SD)	88.70 ± 25.59
Uric acid (mg/dL) (mean ± SD)	4.97 ± 1.22
Corrected calcium for serum albumin (mg/dL) (mean ± SD)	9.13 ± 0.77
Phosphate (mg/dL) (mean ± SD)	4.26 ± 1.13
Sodium (mEq/L) (mean ± SD)	139.84 ± 2.98
Potassium (mEq/L) (mean ± SD)	4.70 ± 0.69
Chloride (mEq/L) (mean ± SD)	106 ± 4.33
CRP (mg/L) (mean ± SD)	14.24 ± 57.29
Hemoglobin (g/dL) (mean ± SD)	9.5 ± 1.35
Leukocytes (×10^9^/L) (mean ± SD)	7.03 ± 2.30
Platelets (×10^9^/L) (mean ± SD)	198.53 ± 62.55
AVF as a venous access, *n* (%)	24 (55.8%)
Hemodiafiltration as KRT, *n* (%)	32 (74.4%)
Average interdialysis weight gain (cc) (mean ± SD)	1622.5 ± 1018.67
Kt/V (mean ± SD)	1.7 ± 0.5
Residual diuresis (cc) (mean ± SD)	2097.67 ± 765.08
Inflammed patients defined as CRP ≥ 2 mg/L, *n* (%)	23 (67%)

**Table 2 ijms-25-09364-t002:** Numerical summary of metabolomic variables with significant differences in their medians based on inflammation = no and inflammation = yes.

Variable	Median Inflammation = No	IQR (Interquartile Range) Inflammation = No	Median Inflammation = Yes	IQR Inflammation = Yes	Mann–Whitney Statistic	*p*-Value
Indole-3-acetaldehyde	1,190,130	579,521	778,141	467,437	316	0.0023
Tryptophan	2,988,287	2,263,933	2,191,984	1,182,744	313	0.0031
Methionine	937,980	238,906	641,991	319,903	310	0.0041
Benzoic acid	42,742	18,539	29,556	14,609	309	0.0045
Carnitine	8,222,7848	73,058,260	58,997,524	53,464,650	295	0.0145
2-Ketobutyric acid	872,432	494,891	517,984	433,389	291	0.0196
Alanine	7,982,122	1,829,652	6,727,464	1,840,257	286	0.0281
Urocanic acid	15,909	15,612	8828	13,043	284	0.0323
2-Methyl-3-ketovaleric acid	65,164,816	25,911,640	51,769,852	22,772,580	284	0.0323
Trehalose	74,993	45,349	100,659	51,935	123	0.0370
D-2-Aminobutyric acid	467,418	132,337	379,133	248,834	282	0.0370
N-Acetyl-L-phenylalanine	50,225	36,626	70,619	56,563	124	0.0396
Tyrosine	2,760,424	852,724	2,470,091	1,027,556	280	0.0423
N-Acetylcarnitine	31,762,998	18,670,840	26,701,374	14,508,230	278	0.0481

**Table 3 ijms-25-09364-t003:** Variables, standardized coefficients, and odds ratios obtained (LASSO).

Variables	Standardized Coefficients	Odds Ratio
Allantoin	1.1695281	3.2204726
Benzoic acid	−0.9747088	0.3773022
Indole-3-acetaldehyde	−0.7354908	0.4792702
Methionine	−0.7067793	0.4932302
Citrulline	−0.5962893	0.5508519
a-Ketoglutarate	−0.5466127	0.5789074
N-acetyl-L-ornithine	−0.5330003	0.5868416
Taurodeoxycholic acid	0.4326940	1.5414044
3-4-Dihydroxybenzeneacetic acid	−0.4005447	0.6699550
Norepinephrine	0.4000314	1.4918716
Pyroglutamic acid	0.3795375	1.4616085
L-Hydroorotic acid	0.2959347	1.3443824
N-Acetylcarnitine	−0.1682766	0.8451200
3-Indolepropionic acid	−0.1674606	0.8458099
Pantothenate	−0.1585006	0.8534225
Cholic acid	−0.1503721	0.8603877
Guaiacol	0.1375204	1.1474251
4-Pyridoxic acid	−0.1233154	0.8839848
b-Aminoisobutyric acid	0.1005446	1.1057730
3-Methyl-Histidine	0.0429894	1.0439269
O-phosphohomoserine	0.0415483	1.0424235
Ureidosuccinic acid	0.0387375	1.0394976
Creatine	−0.0143093	0.9857926
Xanthosine	0.0020708	1.0020730

**Table 4 ijms-25-09364-t004:** Numerical summary of metabolomic variables in post-session with significant differences in their medians based on inflammation = no and inflammation = yes.

Variable	Median Inflammation = No	IQR (Interquartile Range) Inflammation = No	Median Inflammation = Yes	IQR Inflammation = Yes	Mann–Whitney Statistic	*p*-Value
N-Acetyllysine	5267	9939	1550	3945	248	0.0170
Aminoadipic acid	33,806	29,710	21,197	20,038	244	0.0244
m-Coumaric acid	60,008	50,430	99,180	82,241	101	0.0487

## Data Availability

The data on which the article is based are available as Appendix A.

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
