# Peer review of "Metabolic Pathways Affected in Patients Undergoing Hemodialysis and Their Relationship with Inflammation"

_ijms, 2024, doi:10.3390/ijms25179364_

Round 1

Reviewer 1 Report

Comments and Suggestions for Authors

This study applied semi-targeted metabolomics to explore novel metabolic pathways related to inflammation in patients with hemodialysis. The authors found that metabolites positively associated with inflammation include allantoin, taurodeoxycholic acid, norepinephrine, pyroglutamic acid, and L-hydroorotic acid. Conversely, metabolites showing negative associations with inflammation include benzoic acid, indole-3-acetaldehyde, methionine, citrulline, alpha ketoglutarate, n-acetyl-ornithine and 3-4-dihydroxibenzeneacetic acid.

The comments are listed below:

1. How many patients were included in this study in total? How many patients were in the inflammation no group, and how many patients were in the inflammation yes group? Please add the information in the methods and tables.

2. The metabolites upregulated with a fold change of 1.5 in the inflammatory group compared to the non-inflammatory group were identified as significant. However, the comparisons were made by the medians. How much difference in the metabolite quantities of the patients within the same group? Why do not compare by using the mean with standard deviation? What is the correlation of the quantities of each metabolite to its corresponding CRP value?

3. In Table 2 and Table 4, please define IQR.

4. In line 157, the authors described that the objective was to characterize the metabolic profile observed in one-year dialysis-dependent patients before and after undergoing a hemodialysis session. However, there is no result showing whether the metabolites have significant differences between the pre-session samples and post-session samples.   

5. In conclusion, the authors stated that “Non inflamed patients exhibit preserved autophagy and reduced mitochondrial dysfunction. Additionally, the microbiota, particularly uricase producing bacteria and those metabolizing tryptophan, play critical roles.” However, these findings are relayed on the literature reviews, and not directly proved by real experiments. Therefore, the conclusion is overstated.   

6. The title is not so suitable for this article. According to the discussion, it seems that not all the metabolites found here have toxic effects on health. Thus, “uremic toxins” may not be a proper term to describe these metabolites. Another term “renal replacement therapies” may be mixed up with kidney transplantation. According to the article, the target patients seem to receive only hemodialysis, why do not directly use “hemodialysis” in the title? 

Comments on the Quality of English Language

Some spelling errors and styles can be improved. 

Author Response

Dear reviewer,

Thank you for the detailed review of the manuscript. We will try to address all your suggestions to improve the quality of the paper. Please find the detailed responses attached.

Best regards,

María

Reviewer 2 Report

Comments and Suggestions for Authors

The above manuscript presents the relation between uremic toxins and inflammation in patients undergoign replacement therapies. The subject is very interesting and very important in the management of dialyzed patients.

I have some mentions to make:

- more information about inflammation in hemodialyses and peritoneal dialyzed subjects should be added in introduction

- I consider that a very important part of the results should the analyze in relation with the cause of renal failure

- the Discussion section is too long, very general and very hard to be read. Please rewrite this section underlying the main data in literature in relation with your results.

Comments on the Quality of English Language

I have no comments

Author Response

(The authors gave the same response as above.)

Round 2

Reviewer 2 Report

Comments and Suggestions for Authors

I recommend acceptance in the present form.